# Global mistranslation increases cell survival under stress in *Escherichia coli*

**Laasya Samhita** \* , **Parth K. Raval** , **Deepa Agashe** \*

National Centre for Biological Sciences, Tata Institute of Fundamental Research, Bangalore, India

\* laasyas@ncbs.res.in, laasya2@gmail.com (LS); dagashe@ncbs.res.in (DA)

## Abstract

Mistranslation is typically deleterious for cells, although specific mistranslated proteins can confer a short-term benefit in a particular environment. However, given its large overall cost, the prevalence of high global mistranslation rates remains puzzling. Altering basal mistranslation levels of *Escherichia coli* in several ways, we show that generalized mistranslation enhances early survival under DNA damage, by rapidly activating the SOS response. Mistranslating cells maintain larger populations after exposure to DNA damage, and thus have a higher probability of sampling critical beneficial mutations. Both basal and artificially increased mistranslation increase the number of cells that are phenotypically tolerant and genetically resistant under DNA damage; they also enhance survival at high temperature. In contrast, decreasing the normal basal mistranslation rate reduces cell survival. This wide-ranging stress resistance relies on Lon protease, which is revealed as a key effector that induces the SOS response in addition to alleviating proteotoxic stress. The new links between error-prone protein synthesis, DNA damage, and generalised stress resistance indicate surprising coordination between intracellular stress responses and suggest a novel hypothesis to explain high global mistranslation rates.

**Data Availability Statement:** Whole genome sequences have been uploaded to NCBI and are available here: https://www.ncbi.nlm.nih.gov/sra/PRJNA603153.

## Author summary

Cells make mistakes all the time. Protein synthesis is exceptionally error-prone, which is puzzling because such mistakes are usually harmful. Prior work shows that specific erroneous proteins can be beneficial under specific stresses. However, this cannot explain a high background error rate, because most mistakes will not be useful. We offer a solution to this conundrum: cells that make more mistakes accumulate a key quality-control enzyme, tripping the switch for a broad stress response (the "SOS" response, that repairs damaged DNA). In turn, this increases cell survival. The unexpected link between protein synthesis and DNA damage suggests surprising cross talk across important quality-control processes, and motivates a new hypothesis to explain how and why cells tolerate so many mistakes.

**Funding:** We acknowledge funding and support from the Wellcome Trust-DBT India Alliance-https://www.indiaalliance.org/- (grant IA/E/14/1/501771 to LS and grant IA/I/17/1/503091 to DA); the Indian Council for Medical Research-https://www.icmr.nic.in- (ICMR fellowship 3/1/3/JRF-2015 to PR); the Council for Scientific and Industrial research-https://www.csir.res.in/- (CSIR research grant 37(1629)/14/EMR-II to DA); and the National Centre for Biological Sciences-https://www.ncbs.res.in/- (NCBS-TIFR). The funders had no role in study design, data collection and analysis, decision to publish, or preparation of the manuscript.

**Competing interests:** The authors have declared that no competing interests exist

## Introduction

The rate of protein mistranslation is amongst the highest known error rates in cellular biosynthetic processes, ranging from 1 in 10,000 to 1 in 100 mis-incorporated amino acids in *E.coli* [1,2]. As a result, 10 to 15% of all proteins in an actively growing *E.coli* cell are likely to carry at least one mis-incorporated amino acid [3,4], implying a high tolerance for mistakes. This is puzzling because mistranslation is thought to be deleterious, and cells have evolved several proofreading mechanisms to minimise error [reviewed in 5]. Multiple hypotheses may explain the occurrence of high mistranslation rates. First, high error rates may occur as an unavoidable tradeoff: fast growth requires rapid translation, and proofreading would increase translation times [6–8]. However, other evidence contradicts this hypothesis. For instance, across natural isolates of *E.coli*, mis-incorporation of leucine from a poly-U mRNA chain is not correlated with translation and growth rates [9]. Second, mechanisms that increase robustness (e.g. folding via chaperones and acquiring stabilizing amino acid changes) may minimize the phenotypic impact of mistranslated proteins, weakening selection against mistranslation and allowing high error rates to persist [10,11].

The above hypotheses refer to the generation and maintenance of baseline mistranslation rates. In addition, mistranslation can also increase transiently (over and above baseline levels) under stress. Indeed, a body of work shows that cells elevate mistranslation levels under specific stresses (stress-induced mistranslation) [reviewed in 12,13]. In some cases, this elevated mistranslation is advantageous, because it produces specific mistranslated proteins that are beneficial. For instance, in *Mycobacterium smegmatis*, artificially increasing specific amino acid substitutions at glutamate and aspartate tRNAs generates a mixed population of wild type and mistranslated RNA polymerase molecules [14], imparting greater resistance to rifampicin (an antibiotic that targets RNA polymerase). More generally, high global (rather than protein-specific) mistranslation rates could confer a selective advantage by generating a "statistical proteome"–a bet-hedging strategy whereby a few cells with some mistranslated proteins (rather than all cells with one specific mistranslated protein as in the previous example) can survive an environmental stress [15,16]. Such a mixed proteome can occur both as a consequence of baseline mistranslation (as shown by proteome analyses [1]) or through artificial or stress-induced mistranslation. One example of general, stress-induced proteome-wide beneficial mistranslation comes from work on mis-methionylation in *E. coli*. In anaerobic environments or upon exposure to low concentrations of the antibiotic chloramphenicol, the methionyl tRNA synthetase enzyme (responsible for adding methionine onto target tRNAs) loses its succinyl modifications, increasing survival [17].

Such examples of beneficial induced mistranslation have motivated the idea that a transient increase in mistranslation rates can evolve under positive selection. However, across natural *E. coli* isolates, even baseline mistranslation rates vary at least 10-fold [9], suggesting that error rates do not converge to a single optimum. Could this variability reflect divergent selection for a high level of specific mistranslated proteins, causing increased basal mistranslation rates as a by-product? Mistranslation is typically deleterious due to protein misfolding and subsequent loss of function [reviewed in 13,18]. Thus, the potential benefit of a few specific mistranslated proteins may only rarely outweigh the larger overall cost of global mistranslation. As such, higher baseline mistranslation rates may be selectively favoured only when specific severe stresses are encountered very frequently. It remains unclear whether high basal and non-specific mistranslation rates can evolve under positive selection.

Here, we propose a fresh hypothesis for the persistence of high baseline mistranslation rates, that bypasses the need for specific mistranslated proteins and invokes a fitness benefit of global mistranslation under stress. Our hypothesis relies on prior work showing that specific

kinds of mistranslation activate cellular stress responses. Such stress responses are beneficial for immediate survival in a hostile environment, but are energetically costly; and hence are not constitutively active. In *E. coli*, mistranslation induced by a mutation in the ribosomal protein S4 triggers an RpoS-mediated oxidative stress response [19], and ageing colonies carrying an editing-defective amino-acyl tRNA synthetase activate the SOS response [20]. Similarly, in yeast, increased translation errors activate the unfolded protein response and environmental stress response [21]. Thus, we hypothesize that mistranslation arising from diverse mechanisms could provide an indirect fitness advantage by activating stress responses, without relying on a direct benefit due to specific mistranslated proteins. Prior studies show that in some cases, the beneficial effect of mistranslation in conjunction with stress responses relies on increased mutagenesis [22]. In others, there is a general increase in mutation frequency, with or without an identified benefit [23,24]. Such an increase in mutagenesis can increase the probability of sampling beneficial mutations, but should also impose the associated cost of simultaneously acquiring more deleterious mutations. However, the sampling of beneficial mutations can also be increased by a simple increase in population size. We propose that the activation of stress responses is advantageous because it increases immediate cell survival and prevents a dramatic decline in population size. As a result, the population has a greater probability of sampling beneficial mutations without altering mutation frequency. Such a non-genetic mechanism lends itself to broad generalization, because it does not rely on specific mistranslated proteins; and populations do not incur the costs of constitutive stress responses or an increased genetic load due to deleterious mutations. Depending on the magnitude of the stress and the rate at which it is encountered, this hypothesis may therefore help understand the impact of mistranslation on cellular fitness.

To test our hypothesis, we manipulated baseline mistranslation levels in *E. coli*, both genetically and environmentally. We initiated our study using a strain with genetically depleted initiator tRNA (tRNAi) content (henceforth "Mutant", carrying only one of four wild type "WT" tRNAi genes [25]). As central players in translation, cellular tRNA levels have a major impact on mistranslation rates [26,27], and are rapidly altered in response to environmental change [28–30]. tRNAi levels are especially interesting because translation initiation is a rate limiting step [31], and tRNAi levels change under various stresses. For instance, in *E. coli*, amino acid starvation is accompanied by transcriptional downregulation of tRNAi during the stringent response [32], while mammalian cells reduce tRNAi levels on exposure to stressors such as the toxin VapC [33] and high temperature [34]. Specifically, depleting tRNAi increases mistranslation rates by allowing promiscuous non-AUG initiation by elongator tRNAs [27,33]. We therefore tested whether mistranslation resulting from tRNAi depletion leads to a general survival advantage.

We first carried out a Biolog screen [35] comparing WT and Mutant growth across 24 antibiotics with various modes of action. The Mutant consistently showed higher growth in the presence of Novobiocin (S1 Fig), a fluoroquinolone antibiotic that inhibits DNA gyrase. Further work showed that increasing baseline mistranslation rates via multiple mechanisms conferred protection against several kinds of DNA damage, via induction of the well-studied bacterial SOS response. Increased mistranslation brings cells closer to the intracellular molecular threshold for SOS induction, such that mistranslating cells sense and repair DNA damage sooner than the wild type. The resulting increase in early survival facilitates the eventual emergence of genetic resistance under antibiotic stress. Importantly, decreasing baseline mistranslation levels has the opposite effect. The mistranslation-induced SOS response is also beneficial under other stresses such as elevated temperature. Thus, we have uncovered a novel link between mistranslation and DNA damage that integrates two major cellular pathways and suggests a new role for mistranslation in cellular survival.

## Results

### Mistranslation increases resistance to DNA damage by enhancing early cell survival

A preliminary screen using Biolog plates indicated that the mistranslating Mutant performed better than the WT when exposed to the DNA damaging antibiotic Novobiocin (S1A–S1C Fig). In most other antibiotics included in the Biolog array plate, the Mutant and WT showed similar performance. We therefore further tested WT and Mutant under different kinds of DNA damage. Compared to WT, the mistranslating Mutant with depleted tRNAi showed higher survival under three kinds of DNA damage, imposed by exposure to UV radiation (base dimerization), hydrogen peroxide (base oxidation) or the antibiotic ciprofloxacin ('Cip', a potent DNA gyrase inhibitor that causes double stranded DNA breaks) (Fig 1A–1C). Thus, mistranslation seemed to be linked to surviving DNA damage. We explored the reasons behind this intriguing link, focusing on Cip resistance (i.e. the number of Cip-resistant colonies per unit viable count). While the Mutant has a higher mistranslation rate, it also has a lower rate of steady state protein synthesis because of lower tRNAi content [36]. To test whether the Mutant's enhanced survival arises simply from a reduction in protein synthesis rates, we treated WT with the antibiotic kasugamycin (which inhibits initiation without altering tRNA levels). However, this treatment did not improve WT Cip resistance significantly (S2 Fig). The Mutant also grows more slowly than the WT in rich media (LB) (mean Mutant growth rate = 0.93/h; WT = 1.15/h; S3A and S3B Fig); such slow growth can enhance survival under stress as a by-product. However, decreasing WT growth by culturing it in glycerol (which causes a 5-fold higher doubling time) had no impact on Cip resistance (S3C Fig). Finally, the Mutant did not produce more GyrA protein (subunit A of DNA gyrase, which is the target of Cip) (S4 Fig); hence, this could not explain enhanced Cip resistance.

Whole genome sequencing showed that each Cip-resistant ($Cip^R$) colony of WT and Mutant (after 24 h on Cip plates) had a single mutation within the well-known QRDR (Quinolone Resistance Determining Region) of the *gyrA* gene (Fig 1D). However, as described above, Mutant cultures produced more resistant colonies per unit viable cell. Thus, while WT and Mutant cells both acquired beneficial mutations in *gyrA*, the Mutant was either sampling them more frequently or earlier than the WT, resulting in a higher final number of $Cip^R$ colonies. A standard rifampicin (Rif) resistance assay [20,24] showed that WT and Mutant had similar basal mutation frequencies (Fig 2A), suggesting that mutation frequency could not explain differential Cip resistance. While assessing Cip resistance, we incubated plates for 24 h, as per the standard protocol for *E. coli*. However, prior work showed that an *E. coli* strain with compromised aminoacyl tRNA editing activity had increased mutation frequency, producing more Rif resistant colonies over time(from 60 – 180h of incubation [20]). Thus, we also carried out the MAC (Mutation accumulation in ageing colonies) assay. We did not see any difference in mutation frequency between WT and Mutant even after 144 h of incubation (S5 Fig), emphasizing that the observed survival differences do not rely on mutagenesis. Instead, we found that Mutant cells had higher early survival upon exposure to Cip (after 2 h; Fig 2B), and sequencing confirmed that cells sampled at this point did not have any *gyrA* QRDR mutations. Therefore, this early survival was not due to genetic resistance, but implies a form of tolerance. Together, these results suggested that mistranslation indirectly enhanced Cip resistance by increasing early survival.

To test the generality of this result, we manipulated mistranslation levels by (a) increasing WT mistranslation by adding the non-proteinogenic amino acids canavanine or norleucine to the growth medium [19,37] and (b) reducing global mistranslation via hyper-accurate ribosomes. Point mutations in the ribosomal protein S12 increase the fidelity of codon–anticodon

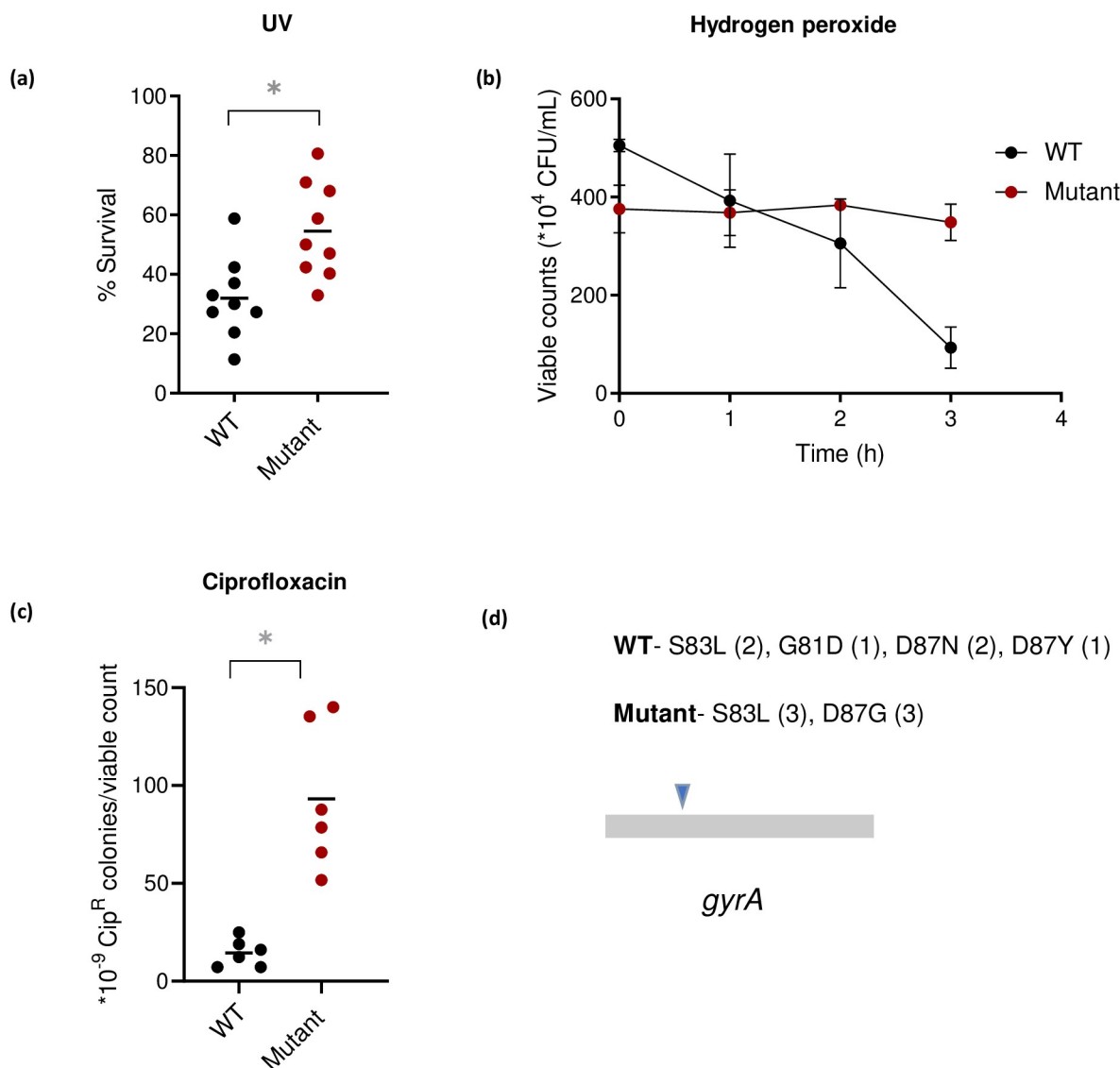

**Fig 1. Mistranslation confers resistance to DNA damage.** (a) Survival of WT and Mutant cultures (n = 9) exposed to 20 J/m$^2$ of UV-C radiation for 5 s. The plot shows mean % survival relative to the number of colonies on a non-irradiated control plate. Mann-Whitney U test, Mutant>WT, U = 9.5, P = 0.0046 (b) Time course of survival (viable counts) of cultures of WT and Mutant (n = 4) treated with 5 mM hydrogen peroxide. At 3 h, Mutant>WT, Mann-Whitney U test, U = 0, P = 0.029 (c) Resistance of WT and Mutant cultures inoculated from single colonies (n = 6), pulsed with 20 ng/mL ciprofloxacin (Cip) for 1 h, and plated on LB agar plates with vs. without 50 ng/mL Cip (Cip-50). The plot shows the average proportion of resistant colonies relative to total viable counts. Mann-Whitney U test, U = 0, P = 0.002. All cells were sampled from mid log phase cultures (OD$_{600nm}$ ~0.6) of the relevant strain. In all figures, asterisks indicate a significant difference. (d) Schematic showing the distribution of Cip$^R$-conferring point mutations in the QRDR (Quinolone resistance determining region) of the *gyrA* gene, found in WT and Mutant upon whole genome sequencing. The number of independent colonies harbouring each mutation is noted in parentheses.

recognition, reducing overall translation errors [38]. To generate hyper-accurate ribosomes, we introduced the mutation K42R, which has almost no impact on growth rate but increases translation fidelity by ~2 fold [39]. Other mutations confer higher fidelity (e.g. K42T) but impose significant fitness costs [39]; hence we did not use them. To estimate the frequency of translational errors due to hyper-accurate ribosomes, we used a modified version of the previously described dual luciferase assay [40] that measures the frequency with which specific

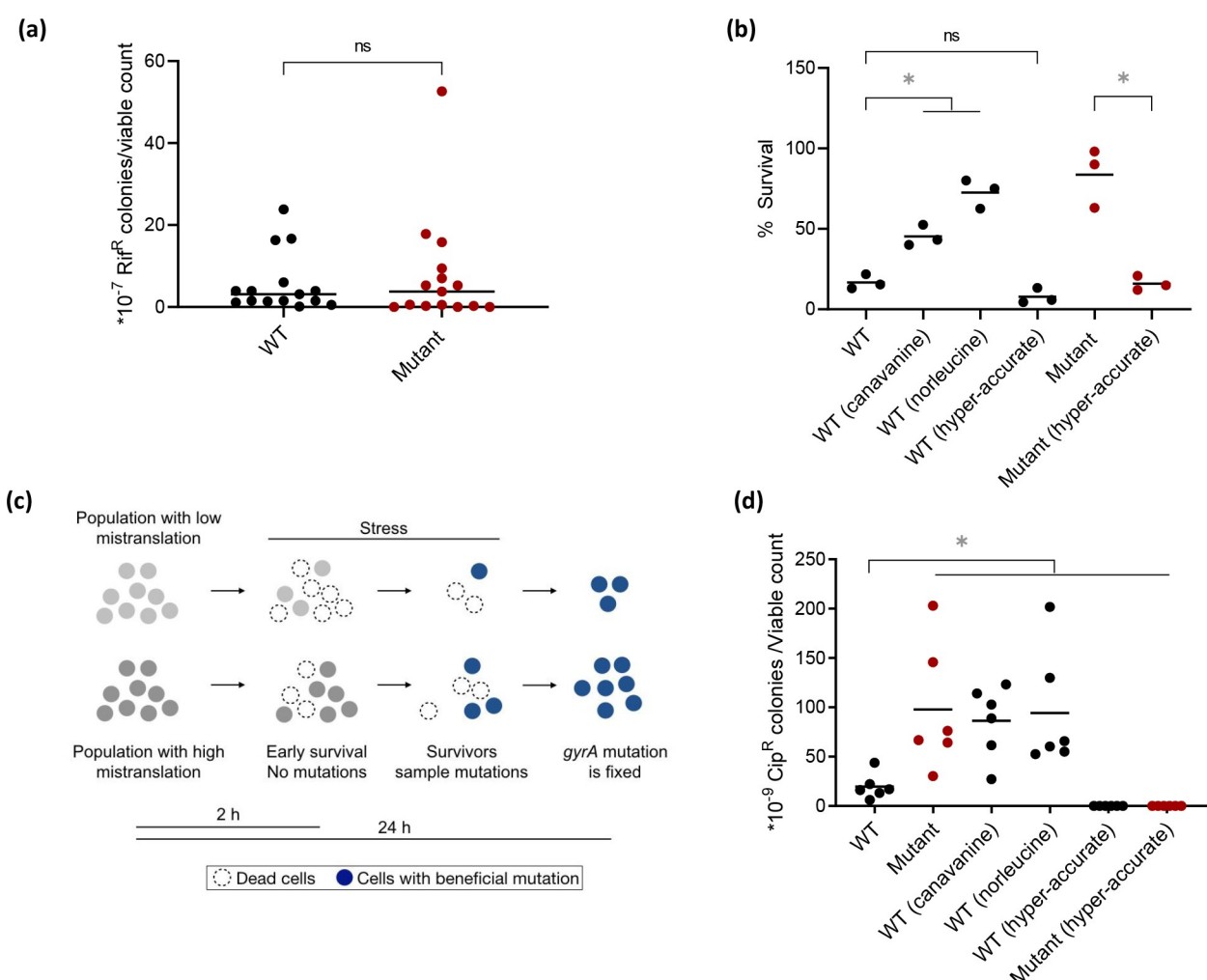

**Fig 2. Mistranslation increases early survival under DNA damage, without altering mutation frequency.** (a) Overnight cultures from single colonies of WT and Mutant were plated on LB agar with 50 μg/mL of rifampicin (Rif-50, n = 15 plates per strain, with paired control plates for viable counts). The plot shows the mean number of resistant colonies per unit viable count from LB plates. Mann-Whitney U test: WT vs Mutant, ns, U = 100.5, P = 0.63. (b) Early survival (tolerance) of WT, mistranslating and hyper-accurate strains treated with Cip-50 for 2 h, from cultures (n = 3) treated with Cip-50 for 2 h. We estimated viable counts before and after exposure. The plot shows mean % survival in each case. t tests: Mutant>WT, t = 6.13, P = 0.02; WT(canavanine)>WT, t = 6.1, P = 0.005; WT(norleucine)>WT, t = 9.6, P = 0.002; WT(hyper-accurate) vs. WT, ns, t = 2.3, P = 0.08; Mutant(hyper-accurate)<Mutant, t = 6.2, P = 0.02 (c) Proposed model for how mistranslation leads to increased sampling of beneficial mutations by enhancing early survival. (d) Resistance of WT, mistranslating and hyper-accurate strains to Cip-50, from cultures (n = 6) pulsed with Cip-20 for 1 h and plated on Cip-50 LB agar. The plot shows the mean proportion of resistant colonies relative to total viable counts. Mann-Whitney U tests: Mutant>WT: U = 1, P = 0.0043; WT(canavanine)>WT: U = 1, P = 0.0043; WT(norleucine)>WT: U = 0, P = 0.0022; WT (hyper-accurate)<WT: U = 0, P = 0.0022; Mutant(hyper- accurate)<Mutant: U = 0, P = 0.0022; Mutant(hyper-accurate)<WT: U = 0, P = 0.0022. Asterisks indicate a significant difference. All cells were sampled from mid log phase cultures (OD$_{600nm}$ ~0.6) of single colonies of the relevant strain, unless indicated otherwise.

codons are misread. We found that both WT and Mutant mistranslation levels decreased ~8 to 10-fold, confirming that the altered ribosomes indeed conferred higher fidelity (S6 Fig). Note that basal misincorporation rates for WT and Mutant are comparable, which is not surprising given that the Mutant is thus far only known to carry out non-AUG initiation [27,33], and no other forms of mistranslation.

With the above manipulations, we found that increasing mistranslation rates consistently increased early survival (Fig 2B) and Cip resistance in WT (Fig 2D); whereas suppressing basal mistranslation decreased early survival and Cip resistance in both WT and Mutant (Fig 2B and

2D). Mutant survival was 5 times higher than WT, and the number of final Cip$^R$ colonies obtained per unit viable count in the Mutant were comparable: ~4.9 times that of WT. This indicates that mistranslation (including basal mistranslation in the WT) indeed increased Cip resistance via an increase in early survival (tolerance), allowing the maintenance of a larger population, and therefore enhancing the number of *gyrA* mutants sampled in liquid cultures that were eventually selected on the Cip plates (Fig 2C). Note that in our whole genome sequencing, we do not see any evidence of increased mutagenesis upon Cip exposure (every colony sampled only had a single point mutation in *gyrA*). Having established this, we next focused on understanding the molecular basis of our finding.

## Mistranslation mediates ciprofloxacin resistance via the SOS response

In response to DNA damage, bacterial cells induce the SOS response, which controls the expression of several DNA repair pathways [41]. Briefly, DNA damage generates single stranded DNA that binds to the protein RecA. Activated RecA stimulates cleavage of LexA (a repressor), which in turn induces the SOS response, de-repressing several DNA repair genes (Fig 3A). When we blocked SOS induction by replacing the WT *lexA* allele with a non-degradable allele [lexA3; 42] and challenged cultures with Cip, both WT and Mutant had lower early survival (Fig 3B) and negligible Cip resistance (Fig 3D), as expected in the absence of an intact DNA repair response. Thus, the mistranslation-induced increase in tolerance leading to Cip resistance depends on the SOS response.

The SOS response has two opposing aspects: rapid DNA repair, and increased mutagenesis due to the activation of error-prone polymerases. The latter temporarily elevates mutation rate, increasing the supply of beneficial mutations [43]. However, as shown above, WT and Mutant had similar basal mutation frequencies (Fig 2A). Therefore, we reasoned that the increased early survival of mistranslating strains must be aided by faster or more efficient repair and recombination. To test this, we deleted RecN–a key member of the SOS-linked recombination mediated repair pathway [44]. The deletion decreased early survival upon Cip exposure (Fig 3C) and led to a complete loss of Cip resistance (Fig 3D), indicating that repair and recombination functions indeed underlie the increased Cip resistance observed in the mistranslating Mutant.

All antibiotics–especially fluoroquinolones such as ciprofloxacin–are thought to act at least in part through the generation of reactive oxygen species (ROS) [45] that damage proteins and increase mutagenesis [24,45,46]. Therefore, we tested whether differential mitigation of reactive oxygen species could explain survival differences in our strains. However, despite prior treatment with an anti-oxidant (glutathione), mistranslating cells continued to show a survival advantage under Cip (S7 Fig). In conjunction with the lack of evidence for increased mutagenesis in our mistranslating strains (Figs 2A, S5 and 1E), this suggests that mistranslation does not alter the role of ROS in Cip toxicity, and hence cannot explain our results.

## Mistranslation enhances Cip resistance by allowing faster induction of the SOS response

Since both WT and Mutant rely on the SOS response for Cip resistance, the ~5-fold greater survival of the Mutant (Fig 1D) continued to be a puzzle. We hypothesized that the survival advantage arose from differential induction of SOS due to mistranslation, allowing rapid DNA repair. Consistent with this hypothesis, greater mistranslation was associated with slightly (though not significantly) higher basal RecA levels across multiple experimental blocks (Fig 4A). However, all mistranslating strains rapidly degraded LexA when exposed to Cip for just 1 hour; conversely, hyper-accurate strains were unable to degrade LexA in this period (Fig 4B).

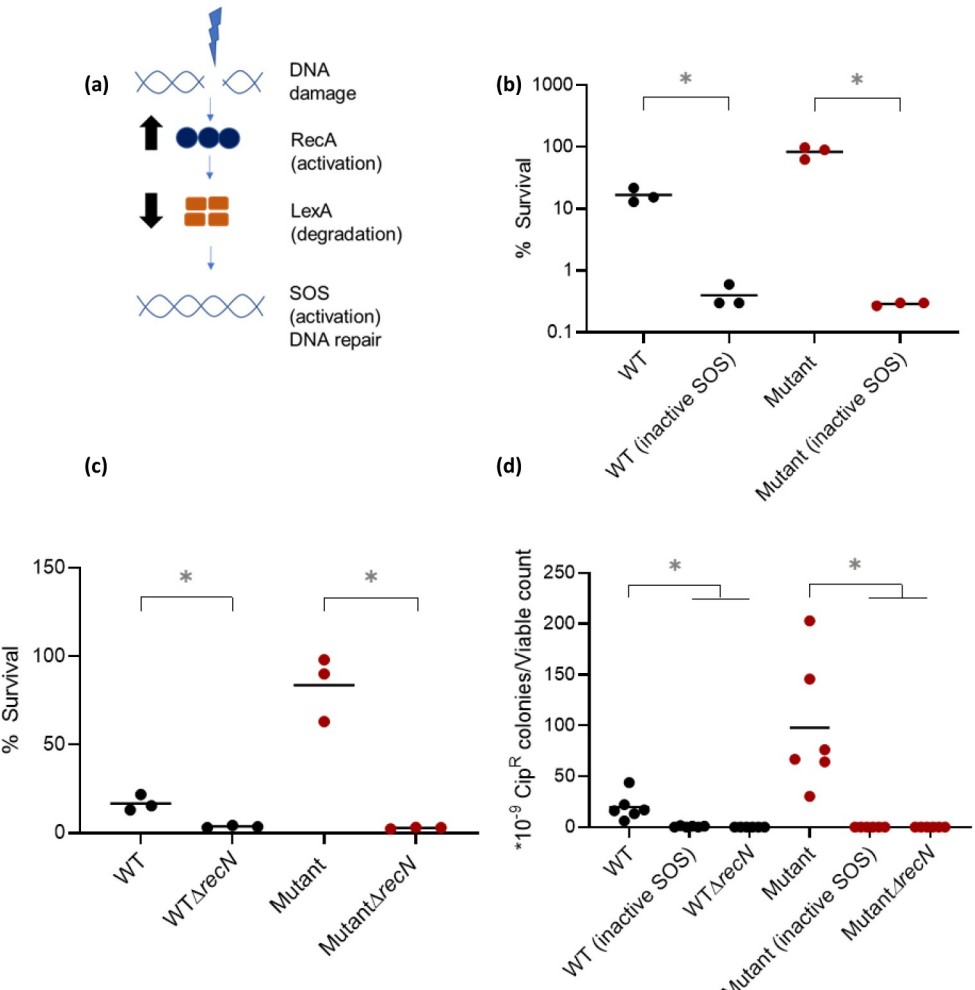

**Fig 3. Mistranslation increases early survival and ciprofloxacin resistance via the SOS response.** (a) Schematic of the SOS response in *E.coli* (b-c) Early survival of indicated strains inoculated from single colonies (n = 3), treated with 50 ng/mL Cip for 2h, and then plated on plain LB. Plots show the mean % survival calculated from total viable counts before and after Cip treatment. (b) Effect of inactivating the SOS response using the *lexA3* allele. Unpaired t test, WT*lexA3* or WT(inactive SOS)<WT, t = 6.2, P = 0.02; Mutant*lexA3* or Mutant(inactive SOS)<Mutant, t = 7.8, P = 0.02; WT (inactive SOS) vs Mutant (inactive SOS), ns, t = 0.96, P = 0.43 (c) Effect of inhibiting recombination, tested by knocking out the *recN* gene. Unpaired t test, W∆*recN* <WT, t = 4.9, P = 0.03; Mutant∆*recN*<Mutant, t = 7.6, P = 0.02; WT∆*recN* vs Mutant∆*recN*, ns, t = 0.96, P = 0.43 (d) Resistance of cultures inoculated from single colonies (n = 6) of indicated strains, pulsed with 20 ng/mL ciprofloxacin (Cip-20) for 1 h, and plated on LB agar with vs. without 50 ng/mL Cip (Cip-50). The plot shows the mean proportion of resistant colonies relative to total viable counts. Mann-Whitney U test: Mutant>WT, U = 1, P = 0.0043; WT(inactive SOS)<WT, U = 0, P = 0.0022; Mutant (inactive SOS)<Mutant, U = 0, P = 0.0022; WT∆*recN*<WT, U = 0, P = 0.0022; Mutant∆*recN*<Mutant, U = 0, P = 0.0022.

These results suggest that even in the absence of DNA damage, RecA was already elevated in mistranslating strains, positioning the cell closer to the SOS induction threshold and leading to rapid degradation of LexA upon exposure to Cip (Fig 4D). To test this, we induced the SOS response in WT and Mutant and monitored the time course of LexA degradation. The Mutant degraded LexA within 10–20 minutes of SOS induction, while the WT took twice as long (Fig 4C and S8 Fig). Similarly, LexA was degraded at lower concentrations of Cip in the Mutant (S9 Fig). Note that the Mutant is not already 'stressed'; it has similar basal LexA levels to the WT (S9 Fig). Thus, the Mutant only starts degrading LexA upon encountering DNA damage.

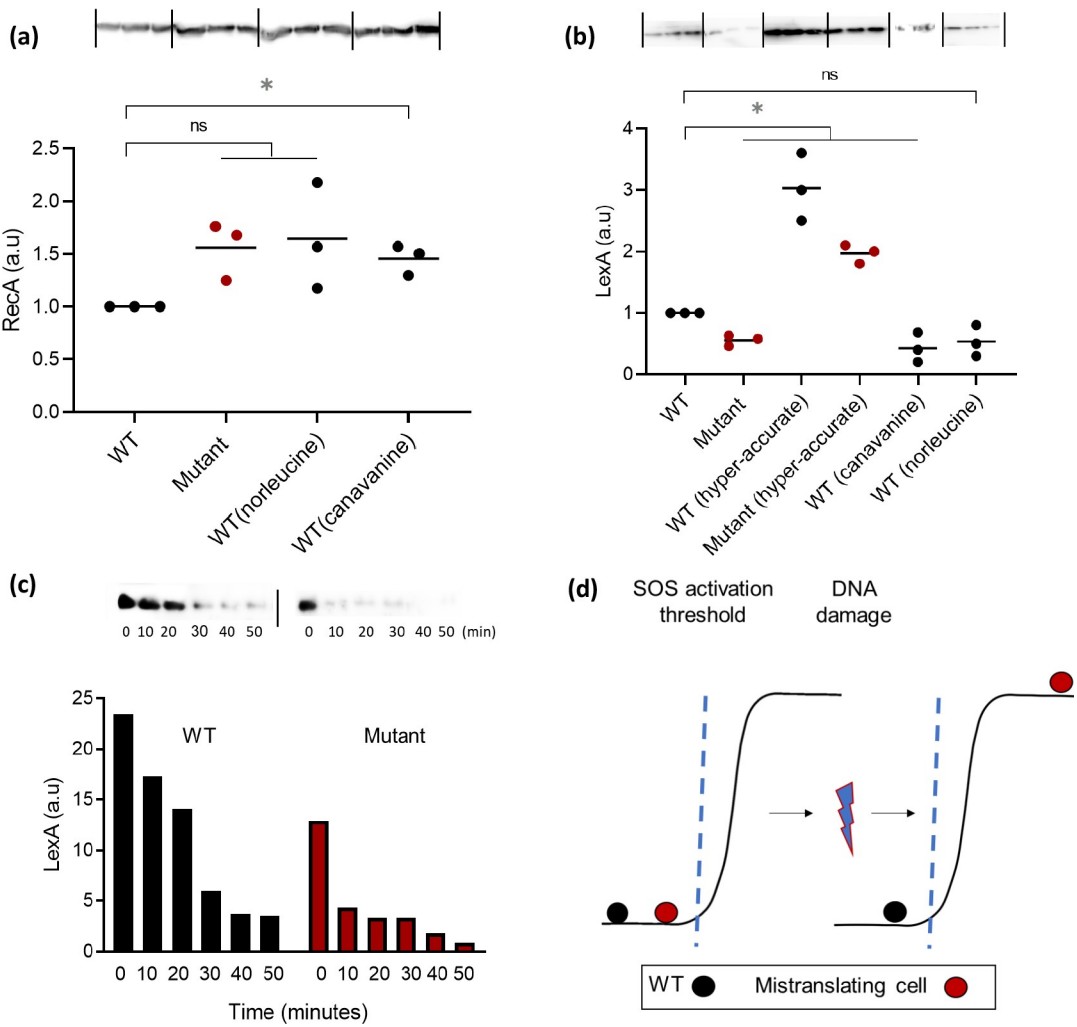

**Fig 4. Mistranslation brings cells closer to the threshold for SOS activation.** (a) A representative blot showing RecA protein levels from mid log phase cultures of WT and mistranslating strains (n = 3). Quantification of mean blot band normalized to total protein is represented in arbitrary units, relative to WT. Paired t tests: WT vs. Mutant, ns, t = 3.5, P = 0.07; WT (canavanine) >WT, t = 5.1, P = 0.03; WT vs WT(norleucine), ns, t = 2.2, P = 0.1 (b) A representative blot showing LexA protein levels in WT and mistranslating strains (n = 3). Quantification of mean blot band normalized to total protein is represented in arbitrary units, relative to WT. Paired t tests: Mutant<WT, t = 8.7, P = 0.01; WT(canavanine)<WT, t = 29.6, P<0.0001; WT vs WT(norleucine), ns, t = 3.4, P = 0.07; WT(hyper-accurate)>WT, t = 12.4, P = 0.006; Mutant(hyper-accurate)> Mutant, t = 10.4, P = 0.009 (c) Time course of LexA degradation from 0 to 50 min after exposure to Cip-20. One blot and bands normalised to total protein are shown here; for more experimental blocks, see S8 Fig. (d) The proposed model of the state of WT and mistranslating strains with respect to the SOS activation threshold.

Together, these observations suggest that in mistranslating strains, (i) LexA is degraded faster upon encountering the DNA damaging stress, and (ii) LexA is degraded at a lower magnitude of the stress (Fig 4D and S9 Fig). Thus, we demonstrate a direct causal relationship between mistranslation, induction of the SOS response, and enhanced survival under Cip.

## Mistranslation activates the SOS response via Lon protease

In our experiments, we increased mistranslation in distinct ways, and consistently observed increased survival under DNA damage. Canavanine and norleucine respectively replace arginine and leucine in the proteome, whereas tRNAi depletion causes mis-initiation with

elongator tRNAs. The parallel outcomes from these diverse modes of mistranslation suggested a general mechanistic link between mistranslation and SOS response. Based on prior studies, we suspected that Lon–a key protease across eubacteria–may represent such a link. In mistranslating *E. coli* cells, Lon alleviates the associated deleterious effects by degrading aggregated and non-functional proteins [3]. In *Pseudomonas*, Lon is essential for RecA accumulation and induction of the SOS response, and is suggested to degrade RecA repressors such as RecX and RdgC [47].

We therefore hypothesized that our mistranslating strains may have higher amounts of Lon, in turn accumulating RecA and bringing cells physiologically closer to the threshold for SOS induction. Because Lon is part of the *E. coli* heat shock regulon [48], we also suspected a general increase in the heat shock response. Indeed, our mistranslating cells had higher levels of Lon protease (Fig 5A), as well as the heat shock transcription factor sigma 32 (S10 Fig; also independently reported recently by [49]). Over-expressing Lon enhanced early survival (Fig 5B) and resistance to Cip (Fig 5C), and reduced LexA levels in both WT and Mutant upon SOS induction (Fig 5D). For technical reasons, we were unable to knock out Lon in our wild type strain KL16. Hence, we deleted Lon in *E. coli* MG1655. While MG1655 had comparable ciprofloxacin resistance to our WT (S11 Fig), deleting Lon decreased Cip resistance (Fig 5C) and increased LexA levels in SOS-induced cells (Fig 5D). Over-expressing Lon also elevated basal RecA levels (in the absence of any DNA damage), further supporting our hypothesis (S12 Fig). Finally, since Lon is part of the heat shock regulon, we predicted that prior exposure to high temperature should induce Lon and increase resistance to DNA damage. True to expectation, cells grown at 42°C for three hours had higher Cip resistance (Fig 6A). Together, these results demonstrate a key role for Lon in mediating Cip resistance by inducing the SOS response.

### Mistranslation-induced SOS response enhances survival in other stresses

As mentioned above, Lon is part of the heat shock regulon; hence we tested whether mistranslation also increased survival under high temperature. As predicted, we found that the Mutant had greater survival at high temperature, especially in the stationary phase of growth (after 12 hours; Fig 6B). We also observed this growth advantage in WT cells treated with norleucine and canavanine, although the results with canavanine were variable (S13 Fig). Importantly, Lon alone could not explain the greater survival at high temperature, which required both mistranslation and a functional SOS response (Fig 6B). Lon levels at 42°C were comparable across WT and Mutant (S14 Fig), and LexA was degraded in the Mutant but not in the WT (S15 Fig). Interestingly, RecN is also important for high temperature survival of both WT and Mutant, suggesting that survival is influenced by functional DNA repair (S16 Fig). The clear dependence of the survival advantage on SOS induction suggests cross-talk between mistranslation, the heat shock response and SOS induction.

Our results thus show that mistranslation-induced early activation of the SOS response is responsible for multiple stress resistance phenotypes. We suggest a model whereby mistranslation of various kinds increases Lon protease levels, triggering an early induction of the SOS response that enhances cell survival not only under DNA damage, but also under other stresses (Fig 7).

## Discussion

High global translation error rates have remained an enduring puzzle, given the large overall costs of generalized mistranslation compared to the benefits of generating specific mistranslated proteins in particular environments. Notably, this problem has mired the hypothesis that

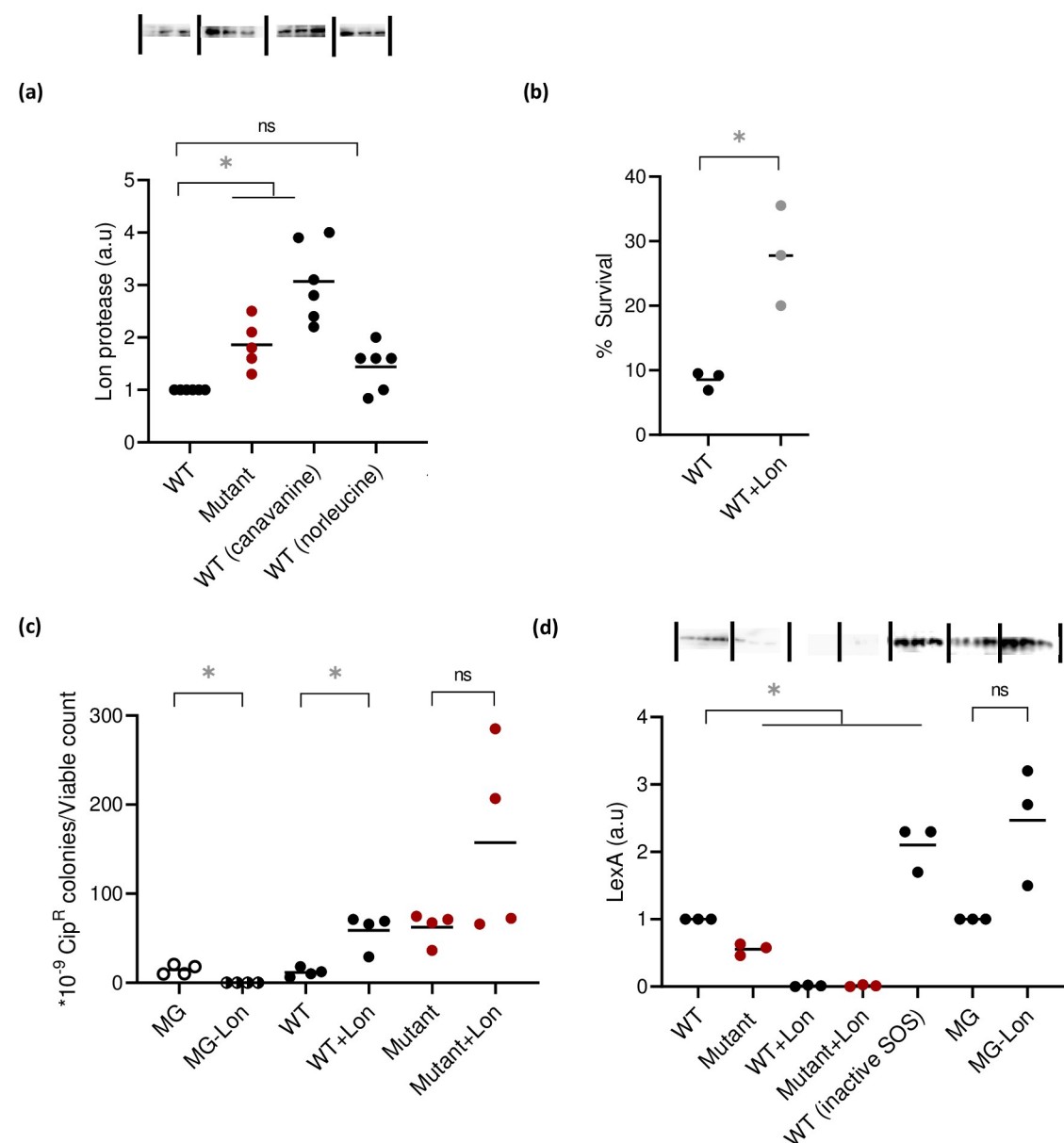

**Fig 5. Mistranslation induces the SOS response via Lon protease.** (a) A representative blot showing Lon protein levels in WT and mistranslating strains (n = 3). Quantification of mean blot band normalized to total protein is represented in arbitrary units, relative to WT. Paired t tests: Mutant>WT, t = 4.2, P = 0.01; WT(canavanine)>WT, t = 6.5, P = 0.003; WT vs. WT(norleucine), ns, t = 2.5, P = 0.05 (b) Early survival of WT and WT+Lon cultures (n = 3) treated with 50 ng/mL Cip for 2 h and then plated on plain LB. The plot shows the mean % survival calculated from total viable counts before and after Cip treatment. Paired t test, WT+Lon>WT, t = 5.1, P = 0.03 (c) Resistance of MG1655, MGΔ*lon* (MG-Lon), WT (KL16), WT+Lon, Mutant and Mutant+Lon on Cip-50 cultures (n = 4) pulsed with Cip-20 for 1 h and plated on LB agar with vs. without Cip-50. The plot shows the mean proportion of resistant colonies relative to total viable counts. Paired t tests: MGΔ*lon*<MG, t = 5.3, P = 0.01; WT+Lon>WT, t = 5.5, P = 0.01; Mutant vs. Mutant+Lon, ns, t = 1.9, P = 0.14 (d) A representative blot showing LexA protein levels in WT and mistranslating strains (n = 3). Quantification of mean blot band normalized to total protein is represented in arbitrary units relative to WT. SOS was inactivated using the *lexA3* allele. Paired t tests: Mutant<WT, t = 8.8, P = 0.01; WT+Lon<WT, t = 171.5 P<0.0001; Mutant+Lon<Mutant, t = 10.4, P = 0.009; WT(inactive SOS)>WT, t = 5.5, P = 0.03; MG vs MGΔ*lon*, ns, t = 3.2, P = 0.08. Asterisks indicate a significant difference between strains. All cells were sampled from mid log phase cultures (OD$_{600nm}$ ~0.6) derived from single colonies of the relevant strain.

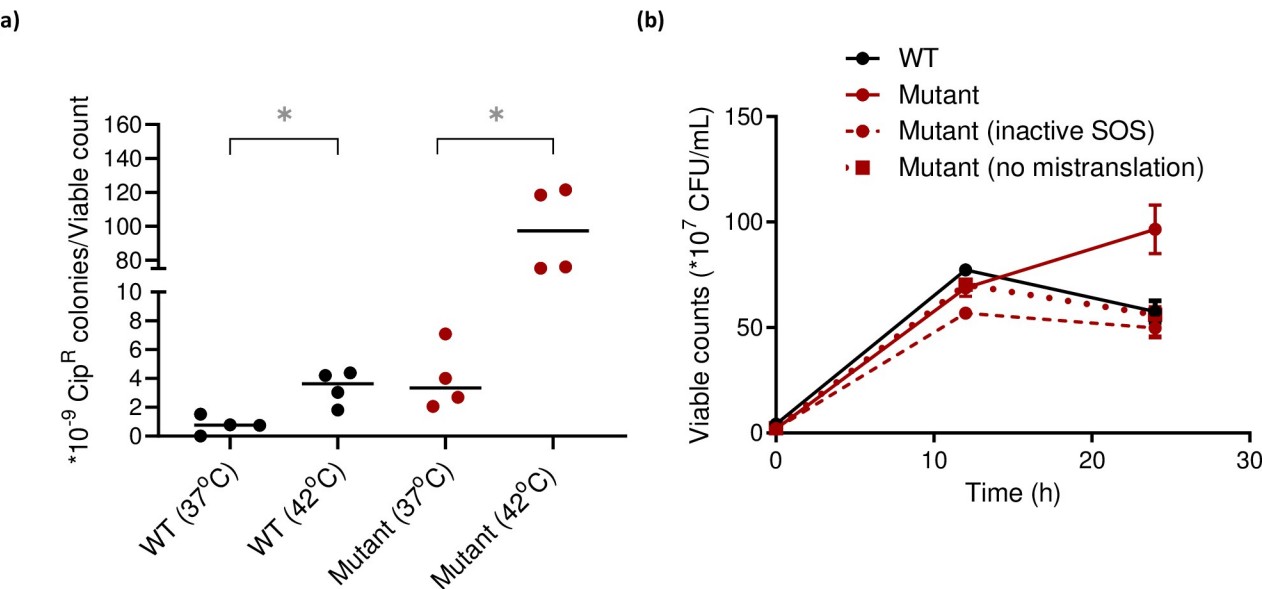

**Fig 6. Mistranslation-induced SOS response enhances survival in other stresses.** (a) Survival of WT and Mutant mid log phase cultures ($OD_{600nm}$~0.6) grown overnight at 37˚C from single colonies (n = 4), sub-cultured and grown at 37˚C or 42˚C (to induce the heat shock response while cells entered log phase) for 3.5 h, and plated on LB agar with vs. without 50 ng/mL Cip (Cip-50). The plot shows the mean proportion of resistant colonies relative to total viable counts. Mann-Whitney tests: WT(42˚C)>WT(37˚C), U = 0, P = 0.0286; Mutant(42˚C)>Mutant(37˚C), U = 0, P = 0.0286 (b) Total viable counts of the indicated strains at 0, 12 and 24 h after exposure to 42˚ C (n = 4, for Mutant (no mistranslation) at 24h, n = 3). SOS was inactivated using the *lexA3* allele. At 24 h, Mann-Whitney tests: Mutant>WT, U = 0, P = 0.0286; Mutant(inactive SOS) vs. WT, ns, U = 4.5, P = 0.4; Mutant(hyper-accurate) vs. WT, ns, U = 8, P>0.99.

mistranslation rates may evolve under positive selection. Here, we diminish this barrier by demonstrating that higher levels of basal mistranslation enhance early survival under DNA damage by allowing rapid induction of the SOS response, making a larger pool of cells available for subsequent genetic change. Note that basal WT levels of mistranslation are critical for

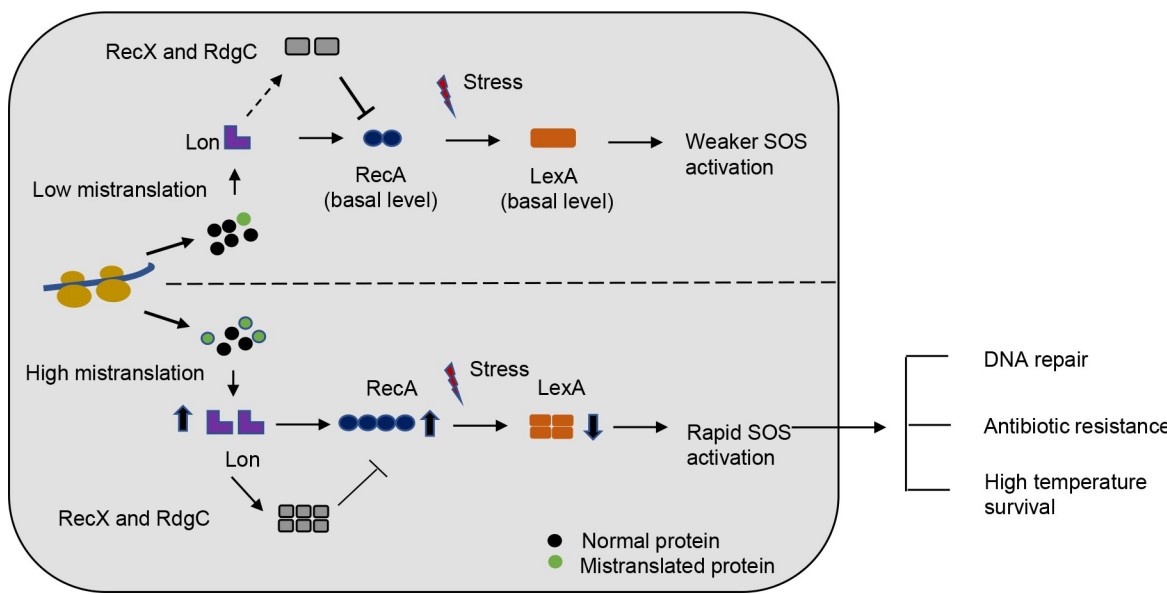

**Fig 7. Proposed model for stress resistance mediated by faster SOS activation in mistranslating strains.**

survival: introducing hyper-accurate ribosomes reduces WT resistance to ciprofloxacin. We further show that this effect is mediated by increased levels of a key protease that deals with aberrant proteins as well as several important regulatory enzymes. Since survival does not rely on the chance generation of specific mistranslated proteins, cells effectively bypass the deleterious load of high mistranslation, triggering a stress response that alleviates damage to cellular components. Most stresses that are commonly encountered by bacteria (oxidative stress, high temperature, radiation, starvation) induce either mistranslation (through damage to proteins), or DNA damage, or both. Hence, it seems fitting that these two phenomena should be linked, and should operate through a common effector molecule. Note that in our mistranslating strains, rapid SOS activation is a by-product of mistranslation, and we do not propose that high mistranslation rates have evolved primarily to activate stress responses. Instead, we propose that the large benefit of mistranslation under diverse stresses (that can be mitigated via the SOS response) may weaken selection against high basal mistranslation. We predict that this should be especially important when stress is encountered frequently or is highly deleterious; this needs to be explicitly tested in further work.

The early tolerance and subsequent high resistance that we observed does not rely on mutagenesis. In contrast, prior work shows that increasing basal mistranslation generates faulty mismatch repair proteins, increasing mutation frequency in *E. coli* [24]. A body of work describing translation stress induced mutagenesis (TSIM) also suggested that altered DNA Polymerase III proteins could underlie the increased mutagenesis observed in strains with mutated *glyV* tRNA anticodons [23]. Lastly, earlier work linking faulty tRNA aminoacylation with activation of the SOS response suggested that the impact of mistranslation on fitness was mediated entirely through elevated global mutagenesis [20]. In contrast to this known mutagenic impact of SOS induction (generating "hopeful monsters"), we demonstrate that mistranslation is generally beneficial under DNA damage because it enhances the rapid DNA repair modules controlled by the SOS response. In our case, this effect is non-specific, acts in the short-term, is induced by different kinds of mistranslation, and leads to increased levels of a key protein quality control molecule. Our proposed model (Fig 7) thus suggests that high global mistranslation could be beneficial under multiple stresses; and could potentially evolve under positive selection. Of course, the generality of this hypothesis needs to be validated with further experiments. More broadly, our results imply that generalized non-genetic changes can facilitate subsequent genetic adaptation by increasing short-term survival. This has been an attractive hypothesis with limited and protein-specific prior support. In *Saccharomyces cerevisiae*, ribosomal frameshifting alters localization of a specific protein, a phenotype that is then fixed by genetic mutations in a few generations [50]. Similarly, changes in Hsp90 levels in *Candida albicans* [51] and at least one phenotype conferred by the prion PSI+ in wild yeasts can be stabilised over evolutionary time [52]. Here, we provide clear evidence that non-directed phenotypic changes can increase stress resistance by enhancing the subsequent sampling of beneficial mutations.

Our work helps to synthesize a diverse body of prior results into a cohesive framework. Mistranslation leads to misfolding and protein aggregation, which is typically deleterious [reviewed in 53]. However, a recent study showed that cells carrying intracellular protein aggregates were more stress-resistant [54]. Interestingly, these cells had upregulated chaperones such as DnaK and proteases such as ClpP, suggesting that protein aggregation can also precipitate stress resistance. Prior work also shows that Lon is required to alleviate the toxic effects of mistranslation [3]. However, thus far only two studies [49] [55] have linked the heat shock response with mistranslation. We now add to this work, clearly showing that the heat shock response is activated by general mistranslation and corroborating prior predictions that

aberrant proteins activate heat shock chaperones [56]. Thus, our work broadly connects mistranslation with the heat shock response, and with increased stress resistance.

Our model could also explain the puzzling observation that Lon protease function determines sensitivity to high concentrations of quinolone antibiotics (which target DNA gyrase, with no direct connection to Lon) such as nalidixic acid [57] and levofloxacin [58]. We show that impairing Lon hinders the ability of cells to rapidly induce the SOS response and repair damaged DNA. The central role of Lon in bridging mistranslation and the SOS response is also supported by previously observed links between the heat shock response and the SOS response. For example, in *Listeria monocytogenes*, heat shock directly triggers the SOS response [59]. In *E. coli*, the heat shock chaperone GroE positively regulates the error-prone DNA Polymerase IV, hitherto thought to be regulated only via the SOS response [60]. Finally, when exposed to levofloxacin, *E. coli* cells express both the SOS response and heat shock genes [58]. These results corroborate our observation that mistranslation and SOS activation together increase heat resistance, though we do not know precisely how this phenotype is regulated. Altogether, we suggest that Lon acts as a key molecule that coordinates several aspects of stress responses, encompassing toxin-antitoxin regulation, survival under anaerobic conditions, SOS, heat shock, antibiotic resistance, and cell division [reviewed in 48].

Finally, our model suggests a general framework to understand the evolution of baseline as well as broad stress-induced mistranslation rates, given the potential generality of the underlying mechanism. We suggest that the "optimal" level of mistranslation rate is set by a fundamental tradeoff between the cost (vs. benefit) of global mistranslation, with the nature of the tradeoff being shaped by the magnitude and frequency of adverse conditions. Thus, mistranslation rates cannot increase indefinitely because they impose a substantial cellular cost under normal conditions (e.g. in our case, the Mutant has ~20% lower growth rate than the WT-S3A and S3B Fig). Similarly, WT cells do not constitutively activate the SOS response because this would lead to a disruption of cell division and increased mutation accumulation [61]. Instead, we suggest a delicate balance whereby cells are poised closer to the threshold for SOS activation; increasing the speed of response once the stress arrives, but not before. DNA damage is a ubiquitous stress for bacterial cells, faced both internally (through the accumulation of reactive oxygen species and replication errors [reviewed in 61,62]) and externally (through UV radiation and more recently, sub-inhibitory antibiotics in soil and water [63]). We speculate that mistranslation levels that extract a cost under normal growth but provide a clear advantage under DNA damage could well be favoured by natural selection. As discussed in the Introduction, generalised mistranslation is already known to activate stress responses. Therefore, the benefit of high mistranslation need not be restricted to DNA damage alone, but could extend to cells facing frequent oxidative stress and anaerobic conditions [19,64]. We therefore suggest that cells could tolerate high mistranslation levels because of the potential benefit under environmental stresses that are encountered frequently.

The SOS response is among the best studied pathways in *E. coli*, inducing DNA repair genes in response to double strand breaks and stalled replication forks generated by severe DNA damage. Yet, we continue to unravel new phenotypes controlled by this response. Instead of being directed solely at DNA repair, the SOS response is turning out to be central for several stressful situations, including high mechanical pressure and antibiotic tolerance [41,65,66]. Similarly, the causes and impacts of mistranslation also continue to be extensively explored, with new details surfacing each day. At the moment, we cannot determine whether mistranslation was co-opted by the DNA damage response as a trigger, or vice-versa. Irrespective of which response evolved first, it is clear that diverse cellular mechanisms are linked in unexpected ways, co-ordinating the cellular response to multiple stresses. While these phenomena are independently well studied, we can now connect them using a single effector

molecule, Lon protease. Our study raises the question of whether such novel links could themselves be evolving in different directions, leading to cross talk between mutation-independent phenotypic variation and genetic change in response to stress.

## Methods

### Strains and growth media

See S1 Text

### Initial screening of WT and mutant strains using phenotype micro arrays

To screen for phenotypic differences between the WT and the mutant across a range of environments, we employed phenotype microarrays. See S1 Text for details.

### Assaying survival under DNA damage

We assessed strain survival under three kinds of DNA damage: exposure to UV light (base dimerization), hydrogen peroxide (base oxidation) and ciprofloxacin (double stranded DNA breaks). See S1 Text for details.

### Dual luciferase assay to assess miscoding

To assess the frequency of translational errors, we used a modified version of a previously described dual luciferase assay. See S1 Text for details.

### Measuring mutation frequency using rifampicin resistance

We estimated mutation frequency in WT and mutant cells using spontaneous mutations conferring rifampicin resistance. See S1 Text for details.

### Measuring protein production

To measure levels of candidate proteins responsible for the observed cellular response to stress, we used western blots with a chemiluminescent detection system. See S1 Text for details.

### Whole genome sequencing

To identify mutations responsible for ciprofloxacin resistance we sequenced whole genomes of the resistant colonies. See S1 Text for details.

## Supporting information

**S1 Text. Supplementary methods.**
(DOCX)

**S1 Fig. Phenotype microarray for WT and Mutant.** (a) WT and Mutant (n = 1 for one plate) were inoculated in the phenotype micro-array antibiotic plate PM12B. Differences in the area under the curve of growth (AUC) for Mutant-WT after 48h are shown here in a heat map. Each of the 24 antibiotics is present in 4 wells, with 2X increasing concentrations in successive wells. Absolute concentrations are proprietary and unknown. (b) Plate showing the antibiotics used. (c) Expanded version showing raw growth curves as obtained from the software, with the arbitrary numbers indicating dye reduction values. Red shows WT dye reduction, green, Mutant, and yellow shows the region of overlap. The four highest values are highlighted

automatically by the software (white rectangles).
(TIF)

**S2 Fig. Reducing WT protein synthesis does not impact Cip resistance.** Survival of WT and WT+Kasugamycin (Ksg) cultures from single colonies (n = 4) pulsed with 20 ng/ml ciprofloxacin (Cip) for 1 hr and then plated on LB plates with vs. without 50 ng/ml Cip (Cip50). Plot shows the number of resistant colonies per unit viable count from LB plates, means are indicated. Mann-Whitney U test, WT vs WT (Ksg), ns, U = 5, P = 0.49
(TIF)

**S3 Fig. Reducing WT growth rate does not impact Cip resistance.** (a) Raw growth curves for WT and Mutant (n = 40) showing mean +/- SD values as obtained by a Tecan growth reader recording $OD_{600}$ every 30 minutes ((b) Mutant grows slower than the WT. WT and Mutant (n = 44) doubling times estimated from the growth curve, means are indicated. Mann-Whitney U test, WT >Mutant, U = 4, P<0.0001 (c) Reducing WT growth rate does not impact ciprofloxacin resistance. Survival of WT and Mutant cultures from single colonies (n = 6) pulsed with 20 ng/ml ciprofloxacin (Cip) for 1 hr and then plated on LB plates with vs. without 50 ng/ml Cip (Cip50). Plot shows the number of resistant colonies per unit viable count from LB plates, means are indicated. Mann-Whitney U test, WT vs WT (glycerol), ns, U = 1, P = 0.07.
(TIF)

**S4 Fig. The increased Cip resistance in the Mutant is not mediated by GyrA upregulation.** Cell extracts from WT and Mutant (n = 3) were used to carry out western blotting for GyrA. Means are indicated. Unpaired t test, WT vs Mutant, ns, t = 0.305, P = 0.76
(TIF)

**S5 Fig. WT and Mutant show similar mutation frequencies in ageing colonies (MAC).** ~1000 cells each from mid-log phase cultures of WT and Mutant (n = 4) were plated onto Rif-50 agar plates and incubated for 5 days. Plot shows the number of rifampicin resistant colonies per unit viable count after this period, means are indicated. Mann-Whitney U test, WT vs Mutant, ns, U = 13.5, P = 0.17
(TIF)

**S6 Fig. Hyper-accurate strains show lower mistranslation.** (a) Schematic showing the Renilla (R-luc) and Firefly (F-luc) luciferase genes along with the correct and incorrect (mistranslating) codon recognition by $tRNA^{Lys}$ (b) Mean mistranslation rates measured with an *in vitro* dual luciferase assay for WT and Mutant (n = 3). Paired t tests, WT (hyper-accurate)<WT, t = 6.9, P = 0.006, Mutant (hyper-accurate)<Mutant, t = 4.8, P = 0.008)
(TIF)

**S7 Fig. Treatment with an anti-oxidant does not eliminate increased Cip resistance due to mistranslation.** Survival of midlog phase cultures ($OD_{600nm}$~0.6) of indicated strains, inoculated from single colonies (n = 3) treated with 50 ng/mL Cip for 2h and then plated on LB agar. The plot shows the average ratio of number of colonies before and after Cip treatment. Unpaired t tests with glutathione treatment: Mutant>WT, t = 3.9, P = 0.02; WT (canavanine)>WT, t = 7.5, P = 0.005.
(TIF)

**S8 Fig. Time course of LexA degradation by WT and mutant.** LexA protein levels normalised to total protein as measured by western blotting using a polyclonal anti-LexA antibody. Each panel shows data for an independent experimental block, in addition to the block shown

in Fig 2E.
(TIF)

**S9 Fig. The mutant activates SOS at lower levels of stress.** (a) LexA protein levels in WT and mutant cultures exposed to different Cip concentrations for 30 mins, measured by western blotting using a polyclonal anti-LexA antibody. Quantitation across biological replicates (n = 3) is shown (mean±SD). (b) RecA protein levels in WT and mutant cultures, measured by western blotting using a polyclonal anti-RecA antibody. Quantitation across biological replicates (n = 3) is shown (mean±SD).
(TIF)

**S10 Fig. Mistranslation elevates the heat shock transcription factor sigma 32.** Mean levels of Sigma 32 protein in mid log phase cultures ($OD_{600}$~0.6) assessed by western blotting using a polyclonal anti-sigma 32 antibody. Quantitation across biological replicates (n = 3) is shown. Paired t tests: Mutant vs. WT, ns, t = 3.1, P = 0.08; WT (canavanine) vs. WT, ns, t = 4.08, P = 0.05; WT(norleucine)>WT, t = 4.5, P = 0.04.
(TIF)

**S11 Fig. WT (KL16) and MG1655 show comparable ciprofloxacin resistance.** Resistance of mid log phase cultures of WT (KL16) and MG1655 ($OD_{600nm}$~0.6) from single colonies (n = 6) pulsed with Cip 20 for 1 h and plated on LB agar with vs. without Cip 50. Plot shows the mean number of resistant colonies per unit viable count from LB agar plates. Mann-Whitney U test, WT vs MG1655, ns, U = 10, P = 0.76.
(TIF)

**S12 Fig. Over-expression of Lon protease elevates RecA levels.** RecA protein levels in mid log phase ($OD_{600}$~0.6) cultures assessed by western blotting using a polyclonal anti-RecA antibody, normalised to total protein. The figure shows bands from two separate blots.t test: WT+Lon>WT, t = 5.13, P = 0.03.
(TIF)

**S13 Fig. Mistranslation provides an early but variable survival advantage under high temperature.** Survival of cultures inoculated from single colonies (n = 3), dilution plated at indicated times on LB agar. Two experimental blocks are shown. Plot shows the total viable counts (mean±SEM). (a) t-tests at 24 h: Mutant>WT, t = 7.2, P<0.001; WT(norleucine)>WT, t = 5.6, P = 0.007; WT vs WT(canavanine), ns, t = 0.12, P = 0.9 (b) t-tests at 24 h: WT vs Mutant, ns, t = 2.8, P = 0.06; WT(canavanine)>WT, t = 3.1, P = 0.02.
(TIF)

**S14 Fig. WT and mutant have comparable Lon levels under heat stress.** Mean levels of Lon protein in mid log phase ($OD_{600}$~0.6) cultures of WT and Mutant (n = 3) as measured by western blotting using a polyclonal anti-Lon antibody, normalised to total protein. Paired t test: WT vs Mutant, ns, t = 0.9, P = 0.12.
(TIF)

**S15 Fig. Mutant degrades LexA under heat stress.** Mean LexA protein levels in mid log phase ($OD_{600}$~0.6) cultures of WT and Mutant (n = 3) as measured by western blotting using a polyclonal anti-Lon antibody, normalised to total protein. Paired t test: Mutant<WT, t = 5.3, P = 0.03.
(TIF)

**S16 Fig. RecN provides an early survival advantage under high temperature.** Survival of cultures inoculated from single colonies (n = 3 for WTΔ*recN* and MutantΔ*recN* and n = 5 for WT

and Mutant), dilution plated at indicated times on LB agar. Plot shows the total viable counts (means±SEM). Unpaired t-tests at 12 h: WT$\Delta$r$ecN$<WT, t = 17.8, P<0.0001; Mutant$\Delta recN$<Mutant, t = 5.2, P = 0.002. At 24 h: WT$\Delta recN$ vs. WT, ns t = 0.79, P = 0.28; Mutant-$\Delta recN$ <Mutant, t = 5.2, P = 0.002.
(TIF)

## Acknowledgments

We thank Anjana Badrinarayanan, Sunil Laxman, and members of the Agashe lab for discussion and critical comments on the manuscript. We thank Awadhesh Pandit and Tejali Naik for help with whole genome sequencing at the NCBS sequencing facility; Dipankar Chatterjee and Kuldeep Gupta (Indian Institute of Science, Bangalore) for access to and help with their Biolog machine; Kurt Fredrick (Ohio State University) for the luciferase assay system; and Jayaraman Gowrishankar and Nalini Raghunathan (Centre for DNA Fingerprinting and Diagnostics, Hyderabad) for the *lexA3* allele.

## Author Contributions

**Conceptualization:** Laasya Samhita, Deepa Agashe.

**Data curation:** Laasya Samhita, Parth K. Raval, Deepa Agashe.

**Formal analysis:** Laasya Samhita, Deepa Agashe.

**Funding acquisition:** Laasya Samhita, Deepa Agashe.

**Investigation:** Laasya Samhita, Parth K. Raval.

**Methodology:** Laasya Samhita.

**Project administration:** Deepa Agashe.

**Resources:** Laasya Samhita, Deepa Agashe.

**Supervision:** Deepa Agashe.

**Validation:** Laasya Samhita.

**Visualization:** Laasya Samhita.

**Writing – original draft:** Laasya Samhita.

**Writing – review & editing:** Laasya Samhita, Deepa Agashe.

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
