## [Decision Letter · Decision Letter 0]

17 Dec 2019

Dear Dr Samhita,

Thank you very much for submitting your Research Article entitled 'GLOBAL MISTRANSLATION INCREASES CELL SURVIVAL UNDER STRESS IN E. COLI' to PLOS Genetics. Your manuscript was fully evaluated at the editorial level and by independent peer reviewers. The reviewers appreciated the attention to an important topic but identified some aspects of the manuscript that should be improved.

They suggest how to improve conceptual framing in the introduction, data presentation and discussion. Importantly, two reviewers ask for the presentation of the results of all 48 phenotypic microarray assays. They consider that this is important to clarify if you proposition that “mistranslation can be adaptive trait” is biased and overstated or not.

We therefore ask you to modify the manuscript according to the review recommendations before we can consider your manuscript for acceptance. Your revisions should address the specific points made by each reviewer.

[LINK]

Yours sincerely,

Ivan Matic

Associate Editor

PLOS Genetics

Kirsten Bomblies

Section Editor: Evolution

PLOS Genetics

Reviewer's Responses to Questions

**Comments to the Authors:**

Reviewer #1: This paper shows that increasing the global rate of mistranslation (in one of a couple of different ways) leads to increased cell survival in several stressful environments due to upregulation of the SOS response. More stress-resistance mutants appear not through mutagenesis but through higher survival yielding more opportunities for mutation. I am not expert in the experimental techniques nor molecular biology details, but it looks sound to me, as do the statistics. I find the manuscript to be novel, important, and convincing.

My only comments/suggestions concern some details of the conceptual framing, primarily in the Introduction. The Introduction lists hypotheses for why mistranslations have evolved to be high, as the question motivating the paper:

1) Speed-accuracy tradeoff

2) Robustness to consequences of error may weaken selection against making them (which is a special case of a broader drift barrier hypothesis that selection isn’t strong enough to care)

3) Bet-hedging advantages of a statistical proteome

4) The proposed new hypothesis of inducing stress responses

An important distinction that the current presentation blurs is that 1) and 2) are hypotheses for the evolution of baseline mistranslation rates, while 4) predicts the evolution of mistranslation rates that rise in response to stress. As currently written, 3) can be read either way, although I think it is a stronger hypothesis if it is with respect to induced high rates. Given that they are evolutionary hypotheses for the causes of somewhat different things, I suggest writing 1) and 2) in one paragraph that is more of a preamble rather than part of a direct numbered contrast to 4), then segueing to inducible vs constitutively high mistranslation rates as part of introducing 3) as real and currently popular hypothesis. Distinguishing between baseline and induced mistranslation rates makes 4) an alternative to 3), but not to 1) and 2) as they are currently written. There are null alternatives re induced rates that are related to 1) and/or 2) but not quite identical, eg that cells in trouble ditch proofreading because they have other stuff to worry about (eg speed in responding to the stress) or that selection is not strong enough to make high fidelity stress-proof, and I think it would be fine to introduce those if made distinct from baseline rates.

Coming back to this framework after the results are presented, what has been shown by the experiments is that it would be beneficial to have higher stress-induced mistranslation rates than those that have in fact evolved. So rather than answer the puzzle of why they have evolved so high, as currently framed in the Introduction, we are left with a puzzle of why they didn’t evolve higher! I suggest acknowledging and maybe speculating about this new puzzle in the Discussion.

Signed,

Joanna Masel

Reviewer #2: This is potentially a very interesting paper. Samhita and et al., examined the effect of higher mistranslation on phenotypes of an E. coli strain and discovered that higher mistranslation rate can cause increase in resistance to a fluoroquinolone antibiotic that inhibits DNA gyrase. They also unveiled the mechanisms of cellular responses to high mistranslation underlying high resistance to fluoroquinolone antibiotics, e.g., SOS response. Subsequently, they showed that high mistranslation rate cause tolerance to high temperature. This is a nice piece of work that convincingly demonstrate that mistranslation trigger SOS response in E. coli, and increase tolerance and resistance to a fluoroquinolone antibiotic and high temperature. The findings presented in this work provide a new knowledge and interest to the audience in PLoS Genetics. However, I have few issues in the manuscript which should be addressed before it is published.

First, I am very confused and frustrated by how the authors presented and discussed about their phenotypic microarray assays of E. coli (WT vs Mutant). Initially, they have done 48 conditions but did not describe what these antibiotics are (no name of antibiotics), but only highlighted three antibiotics that showed what they discussed in the manuscript. This experiment should provide much more meaningful information to this work and the authors should describe carefully in the manuscript and provide all experimental data and information in supplementary figure and table. I don’t see a reason why the authors ignored completely these antibiotics that exhibit negative effects on Mutant. What kind of antibiotics are more negative effects on Mutant over WT? It looks like that the majority of antibiotics are negative on Mutant. Should such negative effects on Mutant provide some information about how mistranslation alters phenotype of E. coli cells too? Can the authors rationalize these negative effects? The colour scheme of Supplementary Fig 1 should be changed, e.g., 0 (WT=Mutant should be white and one colour as positive and another as negative.

Related to the above, I feel that the argument that the author made in the manuscript “mistranslation can be adaptive trait” is overstating. The authors made a strong argument on this subject and generalize their observation e.g., page 10 “we demonstrate that mistranslation is generally beneficial under stress”. The authors only see tolerance to a fluoroquinolone antibiotic and high temperature, and ignored the fact that the most antibiotics cause negative effects on the high mistranslation strain in the manuscript, and it sounds that the authors form a theory based on very biased information that they like to see, and not based on comprehensive and unbiased data. Again, it is hard to discuss here as they did not present the microassay data properly (they should provide full description of their data in the microarray, and discussed the data extensively in the results and discussion), but they author should do more balanced discussion throughout results and discussion. Some recent papers demonstrated recently that (PMID:30936490, PMID: 27111525), genetic optimization of proteins can be understood only considering fitness in multiple environmental effects, but not only one environment. I presume that similar attribute applies to the organismal level. Seeing higher fitness in few environments does not mean that it is generally good, as bacteria are living environments that constantly changing and fluctuating. I suggest the authors to consider such aspect carefully and revise their discussion to more unbiased manner.

It is very unclear how many samples that the authors sequenced (whole genome sequencing) to determine mutations in the genome, and which mutations that they observed. They should provide a complete table and list all mutations observed in the sequenced clones.

Reviewer #3: The manuscript, “Global mistranslation increases cell survival under stress in E. Coli” demonstrates that mistranslation brings cells closer to the threshold level of misfolded proteins required for induction of a stress response. This in turn helps cells survive in stressful environments. This is an exciting finding in the field of evolutionary genetics and cell biology, and has important implications related to medicine and aging. The authors for the most part describe a logical progression through the experiments and how they reached this conclusion. The conclusion is justified by the data.

I do have a few major concerns that need to be addressed.

Major concern 1: does evolution maintain high mistranslation rates globally just because they are beneficial in some stressful environments?

a. The abstract starts by asking why global mistranslation rates are high. I am not sure this manuscript addresses that question. It shows why increased mistranslation rates are beneficial in stressful environments. It does not show they are generally beneficial. Thus, it seems unlikely that evolution maintains high mistranslation rates globally just because they are beneficial in some environments. More problematically, I am not sure the authors are even arguing that this is true. Can they authors make a more explicit statement about whether their results suggest that natural selection maintains high mistranslation rates globally due to the benefit this provides in stressful environments?

b. The authors test 49 different environments using biolog, but only report results from a few in figure 1. What about the other environments? Is mistranslation usually beneficial or deleterious? I bet it is usually deleterious, making it difficult to argue that maintaining high mistranslation rates is adaptive.

c. How come cells don’t just decrease the threshold level required to activate stress responses? Wouldn’t that be more efficient than maintaining sloppy protein synthesis and all of these misfolded proteins?

Major concern 2: Some of the earlier results and experiments are not described in enough detail.

a. See point 1b above. More information is required about these initial experiments including whether the wt strain did better in most environments, and how many replicates were performed.

b. The results section says the mutant outperformed the wt in ‘many’ of the DNA-damaging agents. How many? Which ones? Why not all?

c. Figure 1 shows significant differences between mutant and wt in a subset of conditions. Are the magnitude of these differences very large compared to the way the wt and mutant grow in other conditions? Does this significance account for multiple hypothesis testing (i.e. the fact that 49 experiments were run)? We need to see a summary of the other conditions, or a comment about this is the main text.

d. All of these things should be discussed in the main text in the results section. Including experimental details in the introduction and a ‘see introduction’ in the results section is atypical.

Major concern 3: Explain why stress responses are not activated all of the time.

a. After reading the paragraph of the introduction including line 85, the immediate response is to question why stress responses are not activated all of the time. The reason is that activating these responses slows growth rate a little because these responses are costly to manifest. So it makes sense to activate them only when the situation calls for it, i.e. in stressful environments. Given this manuscript discusses the threshold level of stress that is required to activate these responses, this information needs to be made clearer in the introduction. As written, this section of the introduction is not only confusing, it is also a missed opportunity to get the reader thinking about the idea that organisms make decisions about when to activate their stress responses and these thresholds may differ for different organisms, cells or mutants.

Major concern 4: Improve figure 2C

The circles in a grid imply 96-well plates, or 24-well plates, or some kind of manufactured piece of equipment. If you mean each circle is a cell in a population, perhaps do not place these into a perfect grid. This figure should be larger and more detailed since it is your main hypothesis.

**Have all data underlying the figures and results presented in the manuscript been provided?**

Reviewer #1: None

Reviewer #2: No: See comments in my report

Reviewer #3: No: Figure 1 and S1 do not provide enough information about the relative growth rates of these strains and in which conditions they were surveyed. The blue color on the scale is not very quantitative or helpful. Can they use some colors that are easier to tell apart, or provide numbers rather than colors?

PLOS authors have the option to publish the peer review history of their article (what does this mean?). If published, this will include your full peer review and any attached files.

Reviewer #1: Yes: Joanna Masel

Reviewer #2: No

Reviewer #3: No

---

## [Decision Letter · Decision Letter 1]

24 Jan 2020

Dear Dr Samhita,

Thank you very much for submitting your Research Article entitled 'GLOBAL MISTRANSLATION INCREASES CELL SURVIVAL UNDER STRESS IN E. COLI' to PLOS Genetics. Your manuscript was fully evaluated at the editorial level and by independent peer reviewers. The reviewers appreciated the attention to an important topic but identified some aspects of the manuscript that should be improved.

No additional experiments are required, but we ask you to modify the manuscript according to the reviewers 1 and 3 recommendations before we can consider your manuscript for acceptance. Your revisions should address the specific points made by each reviewer.

[LINK]

Yours sincerely,

Ivan Matic

Associate Editor

PLOS Genetics

Kirsten Bomblies

Section Editor: Evolution

PLOS Genetics

Reviewer's Responses to Questions

**Comments to the Authors:**

Reviewer #1: I am not satisfied with the authors’ response to my previous concern that the writing is unclear about the distinction between baseline mistranslation rates and stress-induced mistranslation rates. In their response, the authors point to lines 55-58 as clarifying that they wish to hypothesize bet-hedging advantages even with respect to baseline rates. But those lines make little sense. Do the authors mean that cells elevate mistranslation above basal rates, rather than that “cells elevate basal mistranslation”, which would seem to me to be an oxymoron? If not, then I cannot parse the sentence (eg do they mean that evolution elevates basal mistranslation?) If they do mean that cells elevate mistranslation above basal rates, then I do not see how this addition makes the hypotheses applicable to baseline levels, as the response to reviews says that it does. The cited data shows that stress-induced elevation actually does happen, and I feel that this fact counts strongly against the authors’ now-stronger-than-before interpretation of the findings of this manuscript as pertinent to baseline rates.

However, I continue to find the results and hence the manuscript as a whole to be novel, important, and convincing. It is a lost opportunity that the conceptualization and framing of the significance and implications of those results has not been improved.

Signed,

Joanna Masel

Reviewer #2: The authors revised according to clarify all of my and other reviewers comments. I don't have any further concerns to publish this manuscript.

Reviewer #3: The authors have addressed my previous concerns about the clarity and lack of detail in the description of the methods and data. A concern that remains is about their hypothesis that there is a “broad fitness benefit of global mistranslation”. The authors frame their paper around this idea, but I am concerned that it is not supported by the data.

The authors show that mistranslation is beneficial in certain environments because it activates cellular stress responses. The link between mistranslation and activation of these responses is interesting. Why not focus the paper there, closer to the data. The argument that this link underlies the baseline mistranslation rates observed across the tree of life is far reaching.

Firstly, understanding whether a trait that is only beneficial in certain environments will be maintained by selection is a complicated question that this paper does not address. It is indeed a difficult question to address because it requires knowledge of how often those environments are encountered, and the selection coefficients across all environments.

Secondly, if activating stress responses is generally a good thing, why wouldn’t these responses evolve such that their baseline level of expression was slightly higher. Thus, mistranslation would not be required to achieve the ideal level of activation.

All reviewers felt that the authors decision to focus on the global fitness benefits of mistranslation was not consistent with their data. Reviewer one asked the authors to separate the evolution of baseline mistranslation rates from the evolution of these rates in high stress environments. Reviewer two writes, “I feel that the argument that the author made in the manuscript “mistranslation can be adaptive trait” is overstating.” I agree with both other reviewers in that the data presented in this manuscript do not tell us anything definitive about the evolution of baseline mutation rates. Nonetheless, I think these data are interesting and the paper should be re-framed to talk about the link between mistranslation and cellular responses to stress. The discussion about how global mistranslation rates evolve could be included in the discussion but should not drive the paper.

**Have all data underlying the figures and results presented in the manuscript been provided?**

Reviewer #1: Yes

Reviewer #2: Yes

Reviewer #3: Yes

PLOS authors have the option to publish the peer review history of their article (what does this mean?). If published, this will include your full peer review and any attached files.

Reviewer #1: Yes: Joanna Masel

Reviewer #2: No

Reviewer #3: No

---

## [Editor Report · Decision Letter 2]

5 Feb 2020

Dear Dr Samhita,

We are pleased to inform you that your manuscript entitled "GLOBAL MISTRANSLATION INCREASES CELL SURVIVAL UNDER STRESS IN E. COLI" has been editorially accepted for publication in PLOS Genetics. Congratulations!

Yours sincerely,

Ivan Matic

Associate Editor

PLOS Genetics

Kirsten Bomblies

Section Editor: Evolution

PLOS Genetics

Comments from the reviewers (if applicable):

**Data Deposition**

http://datadryad.org/submit?journalID=pgenetics&manu=PGENETICS-D-19-01868R2

**Press Queries**

---

## [Editor Report · Acceptance letter]

2 Mar 2020

PGENETICS-D-19-01868R2 

GLOBAL MISTRANSLATION INCREASES CELL SURVIVAL UNDER STRESS IN E. COLI 

Dear Dr Samhita, 

We are pleased to inform you that your manuscript entitled "GLOBAL MISTRANSLATION INCREASES CELL SURVIVAL UNDER STRESS IN E. COLI" has been formally accepted for publication in PLOS Genetics! Your manuscript is now with our production department and you will be notified of the publication date in due course.

With kind regards,

Matt Lyles

PLOS Genetics

On behalf of:
